**Data Availability Statement:** All relevant data are within the manuscript and its Supporting Information files.

# Facilitators and barriers to engagement with contact tracing during infectious disease outbreaks: A rapid review of the evidence

Odette Megnin-Viggars[ID][1]*, Patrice Carter[1], G. J. Melendez-Torres[2], Dale Weston[3], G. James Rubin[4]

**1** Research Department of Clinical, Centre for Outcomes Research and Effectiveness, Educational & Health Psychology, University College London, London, United Kingdom, **2** Peninsula Technology Assessment Group, University of Exeter Medical School, Exeter, United Kingdom, **3** Emergency Response Department Science & Technology, Behavioural Science Team, Public Health England, Porton Down, Salisbury, United Kingdom, **4** Department of Psychological Medicine, Institute of Psychiatry, Psychology & Neuroscience, King's College London, London, United Kingdom

* o.megnin@ucl.ac.uk

## Abstract

### Background

Until a vaccine is developed, a test, trace and isolate strategy is the most effective method of controlling the COVID-19 outbreak. Contact tracing and case isolation are common methods for controlling infectious disease outbreaks. However, the effectiveness of any contact tracing system rests on public engagement. Numerous factors may influence an individual's willingness to engage with a contact tracing system. Understanding these factors has become urgent during the COVID-19 pandemic.

### Objective

To identify facilitators and barriers to uptake of, and engagement with, contact tracing during infectious disease outbreaks.

### Method

A rapid systematic review was conducted to identify papers based on primary research, written in English, and that assessed facilitators, barriers, and other factors associated with the uptake of, and engagement with, a contact tracing system.

### Principal findings

Four themes were identified as facilitators to the uptake of, and engagement with, contact tracing: collective responsibility; personal benefit; co-production of contact tracing systems; and the perception of the system as efficient, rigorous and reliable. Five themes were identified as barriers to the uptake of, and engagement with, contact tracing: privacy concerns; mistrust and/or apprehension; unmet need for more information and support; fear of stigmatization; and mode-specific challenges.

**Funding:** PC and OMV were funded by Go-Science; the review was conducted at the request of the Scientific Pandemic Influenza Group on Behaviours (SPI-B), a behavioural science advisory group for the Scientific Advisory Group for Emergencies (SAGE): Coronavirus (COVID-19) response team, who provide scientific and technical advice to support UK government decision makers. GJR and DW were funded by the National Institute for Health Research Health Protection Research Unit (NIHR HPRU) in Emergency Preparedness and Response, a partnership between Public Health England, King's College London and the University of East Anglia. DW is also supported by the National Institute for Health Research Health Protection Research Unit (NIHR HPRU) in Behaviour Change and Evaluation, a partnership between Public Health England and the University of Bristol.

**Competing interests:** The authors have declared that no competing interests exist.

## Conclusions

By focusing on the factors that have been identified, contact tracing services are more likely to get people to engage with them, identify more potentially ill contacts, and reduce transmission.

## Introduction

The current coronavirus disease 2019 (COVID-19) pandemic represents a major global public health disaster. COVID-19 is caused by the SARS-CoV-2 virus, and is transmitted through respiratory droplets (e.g. coughing or sneezing) or through contact routes (direct contact with an infected person or indirect contact with surfaces or objects that an infected person has made contact with). As of 3rd August 2020, just under 18 million cases had been reported globally, resulting in more than 686,000 deaths [1]. COVID-19 cannot be treated effectively with antiviral drugs and there is currently no vaccine available.

Until a vaccine is developed, a test, trace and isolate strategy is the most effective method of controlling the COVID-19 outbreak [2–4]. This system is heavily reliant on members of the public engaging with it. There are multiple stages where this engagement could be less than optimal, from the decision to report symptoms [5], through to whether people remain in isolation or quarantine for the full period recommended [6]. One critical area that requires specific attention is engagement with the contact tracing element of the system. Different countries approach this in different ways, but the core elements are similar. Contact tracing aims to prevent onward transmission of an infectious disease by identifying, assessing and managing people who have been in close contact with an infected individual. Within England, people who test positive for COVID-19 are contacted by a dedicated service and asked to provide the names and contact details of the people that they live with or have had close contact with recently, as well as any places they have been recently such as a restaurant or workplace. 'Recently' in this context is defined as starting 48 hours before their symptoms began. Armed with this information, the service then attempts to get in touch with these contacts and asks them to enter quarantine, preventing onward transmission of the virus. Unfortunately, there is evidence that some people are declining to provide any details of their contacts to the service or are not providing full details of how to get in touch with them [7]. Deficiencies in the national service have led some local regions to set up their own services, to complement the national endeavour [8].

In addition to manual contact tracing systems, many countries (including the UK) have, or are in the process of, developing digital contact tracing applications. Digital contact tracing systems largely use smartphones to measure the proximity of devices to each other and use this as a proxy for contact between people. These data are then analysed by an algorithm that quantifies risk (using parameters including duration and number of contacts with positive cases) and generates an alert where risk is sufficiently high that action should be taken. However, there are also limitations associated with digital contact tracing systems including: imprecision in detecting contact and distance; vulnerability to fraud and abuse; unestablished effectiveness; reliance on a high level of accuracy of diagnostic testing and a high level of uptake of the application; the need to win public trust and confidence; potentially harmful behavioural impacts; and potential exacerbation of inequalities [9]. In an attempt to improve engagement, some apps may include additional features specifically designed to provide personal benefit, such as

information about current levels of infection and ways to protect oneself, however, the impact of these measures on uptake and engagement is unknown.

Contact tracing is not unique to the COVID-19 pandemic. It has been used extensively in previous emerging infectious disease outbreaks [10]. In this paper, we report a rapid review of contact tracing to identify factors that are associated with greater engagement by patients, defined as greater likelihood of providing full details of all relevant contacts, or of downloading and using an application-based contact tracing system.

## Methods

Following the PRISMA guidelines [11] we developed a protocol for a rapid review to identify factors that influence engagement with contact tracing during major health incidents, unfortunately time restrictions did not allow for the registration of this protocol. This rapid systematic review satisfied all of the PRISMA checklist items.

### Search strategy

A search strategy was developed that included medical subject headings and free text terms. Key words for the search included terms for epidemics/pandemics (including coronavirus, avian influenza, Ebola, Middle East respiratory syndrome, severe acute respiratory syndrome, swine flu) and terms for contact tracing and isolation/quarantine.

Four electronic databases (MEDLINE, Embase, PsycINFO, ProQuest (Coronavirus Research Database, Public Health Database, Social Science Database, Sociology Database and Internal Bibliography of the Social Science [IBSS]) were searched from inception to July 2020. Additionally the pre-print database, Medrxiv was searched on the 15th July 2020. The full list of search terms can be found in S1 Table.

### Selection criteria

To be included in this review, studies had to: report on primary research; be written in English; include factors associated with contact tracing and include participants with experience of a major health incident. Studies were excluded if the major health incident was not viral and contagious, and if the disease was sexually transmitted.

Citations from each database search were downloaded into EndNote and duplicates removed. Titles and abstracts of identified studies were independently screened by two reviewers for inclusion against criteria, and good inter-rater reliability was observed (percentage agreement =>90%). All primary-level studies included after the first screen of citations were acquired in full and re-evaluated for eligibility at the time data was being entered into an Excel-based study database (see S2 Table for studies that were excluded at full-text review).

### Data extraction and synthesis

Data extraction and synthesis were conducted by two reviewers, and discrepancies with coding were resolved through discussion. Quality of each included study was assessed using the CASP (Critical Appraisals Skills Programme) Qualitative checklist for qualitative studies [12] or the BMJ Critical appraisal checklist for survey studies [13]. A quality rating was assigned to each study, where ++ indicates that most (≥75%) or all of the checklist criteria have been met, + indicates that the checklist criteria have been partially met (≥50%-75%), and—indicates that the majority of checklist criteria have not been met (<50%). Thematic synthesis and thematic network analysis [14, 15] was used to combine the results of included studies. Emerging themes were derived from the data presented within the included studies, and were placed into

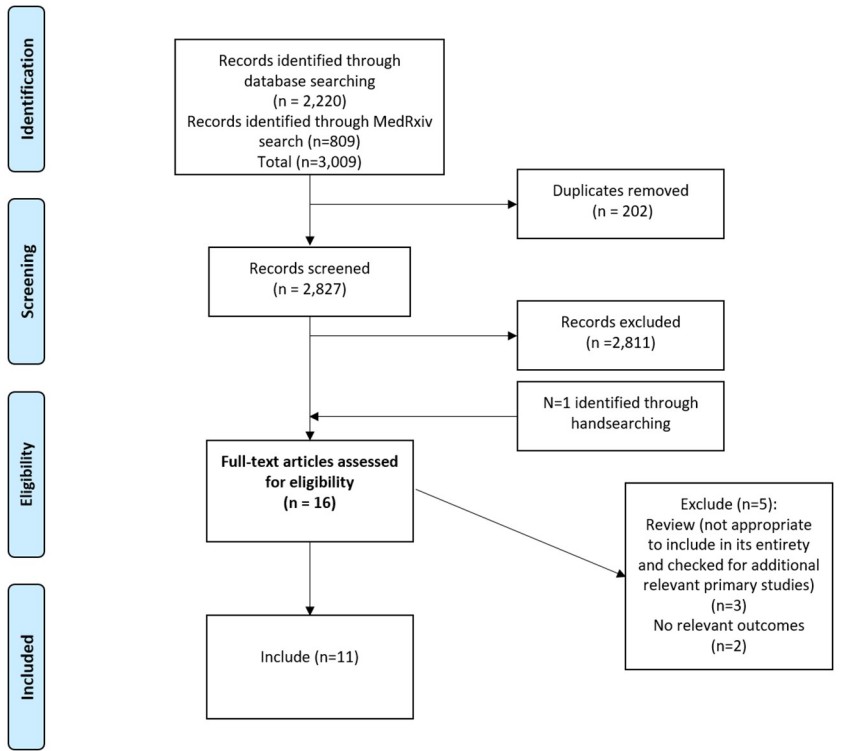

**Fig 1. PRISMA flowchart.**

a thematic map representing the relationship between themes, subthemes and overarching categories, in order to inductively identify common themes across studies.

## Results

The systematic search of electronic databases and the MedRxiv pre-print server generated a total of 3,009 references, after removal of duplicates 2,827 relevant abstracts were assessed for eligibility. Of these, 17 papers were reviewed in full-text (including two papers that was identified through hand searching). Twelve studies met the eligibility criteria and were included in the review (see Fig 1 for flow chart of study search and selection process).

### Included studies

The characteristics of the included studies are summarised in Table 1. Six of the included studies investigated people's attitudes to contact tracing systems for COVID-19 [16–21], five explored people's experience of contact tracing during Ebola outbreaks [22–26], and one study investigated hypothetical scenarios including scabies, shigella and mumps [27]. Five studies included participants with direct experience of contact tracing (with participants including contact tracers, community leaders, healthcare providers, and selected households living in affected/unaffected communities), one study examined experiences of using a pilot contact tracing app (participants were healthcare workers), and six studies explored the acceptability of digital (predominantly application-based) contact tracing systems (with participants including the general population and public health professionals).

**Table 1. Characteristics of included studies.**

| Author, country and reference | Study design | Infectious disease | Participants | Study aim | Sample size | Percentage female | Percentage BAME (non-white) | Included Themes | Study quality[a] |
|---|---|---|---|---|---|---|---|---|---|
| Altmann 2020<br><br>France, Germany, Italy, UK & US | Online—quantitative survey | COVID-19 | General population | To measure the acceptability of an app-based contact tracing system | 5, 995 | 52.3 | NR | • Facilitators of contact tracing<br> • Collective responsibility<br> • Personal benefit<br>• Barriers to contact tracing<br> • Privacy concerns<br> • Mistrust and/or apprehension<br> • Mode-specific challenges | + |
| Bachtiger 2020<br><br>UK | Online—quantitative survey | COVID-19 | Individuals with a previous healthcare event or encounter | To measure the determinants of willingness to participate in an app-based NHS[b] contact tracing programme | 9,512 | 57.0 | NR | • Facilitators of contact tracing<br> • Personal benefit<br> • Co-production of contact tracing systems<br>• Barriers to contact tracing<br> • Privacy concerns<br> • Mistrust and/or apprehension<br> • Unmet need for more information and support<br> • Mode-specific challenges | + |
| Barker 2020<br><br>Liberia | Qualitative (in-depth interviews and focus groups) | Ebola | Health care providers | To understand views on facilitators and barriers to health system resilience and links to community engagement in a localised context | 92 interviews and 16 focus groups | NR | NR | • Facilitators of contact tracing<br> • Co-production of contact tracing systems | - |
| Caleo 2018<br><br>Sierra Leone | Qualitative interviews | Ebola | Household members and community informants | To explore transmission dynamics and community compliance with control measures | 38 | 52.7 | NR | • Facilitators of contact tracing<br> • Collective responsibility<br>• Barriers to contact tracing<br> • Privacy concerns<br> • Mistrust and/or apprehension | + |
| Filer 2020<br><br>UK | Online—quantitative survey | COVID-19 | NHS Staff | To assess uptake, use and difficulties encountered using an NHS contact tracing app | 462 | NR | NR | • Barriers to contact tracing<br> • Unmet need for more information and support<br> • Mode-specific challenges | - |

(*Continued*)

**Table 1.** (Continued)

| Author, country and reference | Study design | Infectious disease | Participants | Study aim | Sample size | Percentage female | Percentage BAME (non-white) | Included Themes | Study quality[a] |
|---|---|---|---|---|---|---|---|---|---|
| Greiner 2015<br><br>Sierra Leone, Guinea, Liberia, Malia Senegal, & Nigeria | Qualitative interviews | Ebola | CDC[c] staff | To explore challenges encountered by CDC staff assisting West African ministries of health to conduct contact tracing | 12 | NR | NR | • Facilitators of contact tracing<br> • Collective responsibility<br> • Personal benefit<br> • Co-production of contact tracing systems<br> • Perception of system as efficient, rigorous and reliable<br>• Barriers to contact tracing<br> • Mistrust and/or apprehension<br> • Fear of stigmatization<br> • Mode-specific challenges | - |
| Helms 2020<br><br>Netherlands | Mixed methods study (qualitative interviews and online quantitative survey) | Scabies, mumps and Shigella | Public health professionals | To assess anticipated advantages and challenges of online respondent driven (online-RDD), and intention to apply online-RDD for contact tracing | 12 interviews and 70 online survey responses | 67.7 | NR | • Facilitators of contact tracing<br> • Perception of system as efficient, rigorous and reliable<br>• Barriers to contact tracing<br> • Mistrust and/or apprehension<br> • Unmet need for more information and support<br> • Mode-specific challenges | ++ (qualitative interviews) - (online survey) |
| Ilesanmi 2015<br><br>Sierra Leone | Qualitative interviews | Ebola | Contact tracers | To identify challenges faced by contact tracers | 12 | NR | NR | • Facilitators of contact tracing<br> • Co-production of contact tracing systems<br>• Barriers to contact tracing<br> • Mistrust and/or apprehension<br> • Mode-specific challenges | - |
| Jansen-Kosternick 2020<br><br>Netherlands | Online quantitative survey | COVID-19 | General population | To identify acceptance of a mobile application for COVID-19 symptom recognition monitoring and contact tracing | 238 | 59.2 | NR | • Facilitators of contact tracing<br> • Collective responsibility<br> • Personal benefit<br>• Barriers to contact tracing<br> • Privacy concerns<br> • Mistrust and/or apprehension<br> • Mode-specific challenges | + |

*(Continued)*

**Table 1.** (Continued)

| Author, country and reference | Study design | Infectious disease | Participants | Study aim | Sample size | Percentage female | Percentage BAME (non-white) | Included Themes | Study quality[a] |
|---|---|---|---|---|---|---|---|---|---|
| Olu 2016 Sierra Leone | Qualitative interviews | Ebola | Contact tracers, and contact tracing supervisors | To understand the characteristics, effectiveness and challenges of contact tracing in Waa and to propose appropriate recommendations for improving contact tracing during future outbreaks. | 10 | NR | NR | • Facilitators of contact tracing • Co-production of contact tracing systems • Barriers to contact tracing • Mistrust and/or apprehension • Fear of stigmatization | - |
| Thomas 2020 Australia | Online quantitative survey | COVID-19 | General population | To explore a) reasons for choosing not to download a mobile application and b) Australians understanding about the app's purpose and capabilities | 1500 | 50.0 | NR | • Facilitators of contact tracing • Personal benefit • Perception of system as efficient, rigorous and reliable • Barriers to contact tracing • Privacy concerns • Mistrust and/or apprehension • Unmet need for more information and support • Mode-specific challenges | - |
| Williams 2020 UK | Online qualitative focus groups | COVID-19 | General population | To explore attitudes towards the UK proposed contact tracing mobile application | 22 | 45.0 | 18.0 | • Facilitators of contact tracing • Collective responsibility • Personal benefit • Barriers to contact tracing • Privacy concerns • Mistrust and/or apprehension • Unmet need for more information and support • Fear of stigmatization • Mode-specific challenges | ++ |

[a] Study quality: Assessed using relevant tool; CASP for qualitative studies and BMJ survey checklist for quantitative surveys: Scores = ++ most of checklist criteria met; + some of checklist criteria met;—insufficient checklist criteria met.

[b]NHS: National Health Service

[c]CDC: United States Centres for Disease Control and Prevention

## Quality assessment of included studies

Five studies used qualitative methods to investigate views/experiences of contact tracing, five studies were survey-based, and two studies used mixed methods. Only two of the included studies were considered high quality, meeting most of the quality assessment criteria [21, 27],

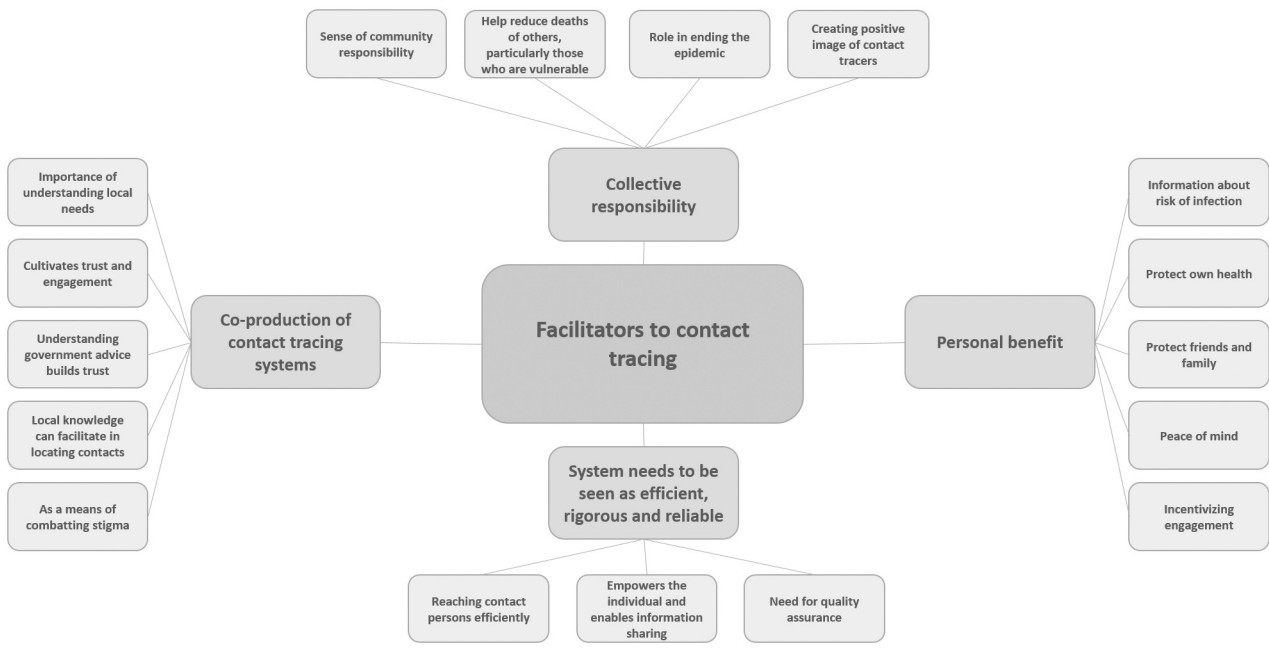

**Fig 2. Facilitators of contact tracing theme map.**

four studies met some of the checklist criteria [16, 17, 19, 22] while six met very few criteria [18, 20, 23–26]. One study included both a qualitative study that met most of the assessment criteria; however, the online survey section of this study met very few of the criteria [27]. See S3 Table for quality appraisal of individual studies.

## Identified themes

The review identified four themes that acted as facilitators to contact tracing: collective responsibility; personal benefit; co-production of contact tracing systems; and the perception of the system as efficient, rigorous and reliable. The review also identified five barriers to contact tracing: privacy concerns; mistrust and/or apprehension; unmet need for more information and support; fear of stigmatization; and mode-specific challenges. See Figs 2 and 3 for theme maps of facilitators and barriers to contact tracing. These themes and their sub-themes are explored in detail below.

## Facilitators of contact tracing

**Collective responsibility.** Participants reported that their intentions to use a contact tracing app were strongly influenced by a sense of collective responsibility [16, 19, 21], and their desire to help reduce the deaths of others, particularly those who are vulnerable [16, 19]. Many viewed contact tracing as a means of ending an epidemic, and embraced their role in this [16, 22] even where participants had some concerns over using a contact tracing app they viewed it as the "only way out" and this collective responsibility was prioritised over personal doubts [21]:

*"I would really support it, I know privacy is really important . . . anything that would help, it doesn't make sense why people wouldn't participate; people are dying all over the world,*

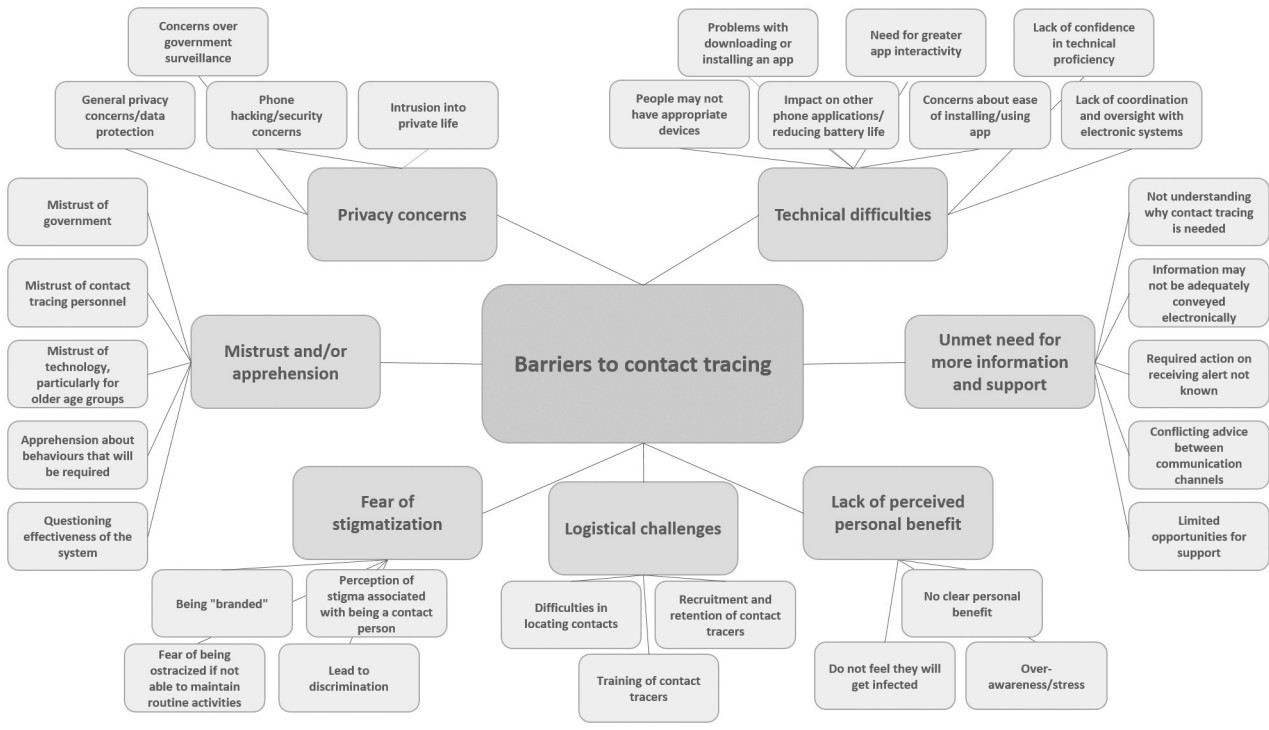

**Fig 3. Barriers to contact tracing theme map.**

*what's more important at the moment, to try and stop this or people knowing what's happened on your phone"*

(Williams 2020; pg. 15)

Some participants described how creating a positive image of contact tracers ("as heroes") had helped to emphasise collective responsibility [23].

**Personal benefit.** Although collective responsibility emerged as a prominent theme in motivating uptake of contact tracing apps, participants also emphasised personal benefit as a facilitator of engagement with contact tracing. Participants were positive about the potential of a contact tracing app to provide information about the risk of infection, and help them gain more insight into the symptoms and spread of the COVID-19 virus [16, 19]. Participants were also motivated by being able to protect their own health [16, 19] and the health of their family and friends [16], and this was particularly compelling for those who feared the virus [19]. Moreover, participants emphasised the peace of mind that a contact tracing app could convey [16, 19].

Conversely, a prominent reason for not intending to use a contact tracing app was that there was no perceived personal benefit which led people to doubt their usefulness [16, 17, 19–21]: some participants felt that existing interventions (such as social distancing) were sufficient and rendered the app unnecessary [20]; some did not believe they would get infected [16, 17]; for others there was the potential for an app to create an over-awareness of the risks and increase stress [19]; and some questioned the acceptability of a contact tracing app in the UK which was perceived as less collectivist and more sceptical of state intervention than other countries which have more widely implemented contact tracing [21].

*"The issue here is will society take it on board and actually do it. I think one of the reasons why places like China, South Korea, Singapore, those Asian countries can successfully manage those situations is because they have a relatively compliant society, people tend to work together or maybe its just because they are used to having their civil liberties curbed to a degree"*

(Williams 2020; pg. 14)

The results showed that where personal benefit may not be initially perceived, uptake could be increased with specific measures; for example, participants who had conducted contact tracing during the Ebola outbreak reflected on lessons learnt and highlighted that financial support provided to contacted persons was used as a means of incentivizing engagement [23].

**Co-production of contact tracing systems.** Participants highlighted the importance of healthcare systems partnering with communities in order to understand local needs [23–26] which was crucial in enabling implementation of contact tracing, and cultivating trust and engagement [23] during the Ebola epidemic. The potential benefit of co-production as a means of building understanding and trust is also highlighted by the association between understanding of government advice and intention to use a contact tracing app [17]. Participants reflected on how community co-production of contact tracing systems can help in a practical way, as local knowledge can facilitate in locating contacts [23], but can also help with acceptance of contact tracing if community engagement can be used as a means of combatting stigma around contact tracing [23, 24].

**Perception of the system as efficient, rigorous and reliable.** Participants considering the advantages of digital contact tracing systems identified the capability of reaching contact persons efficiently and effectively as a potential benefit [20, 27]. Participants were also positive about how digital systems can empower the individual as 'holder' and 'sharer' of their own anonymised data [27]:

*"I think you can take away many barriers by having the index [patient] forward this [the online contact tracing questionnaire]. Especially if it is possible to do so anonymously. . .."*

(Helms 2020, pg. 10)

## Barriers to contact tracing

**Privacy concerns.** A prominent barrier to using a contact tracing app was concern over government surveillance. Participants were worried that their personal information would be used by government to keep watch on them during and after the pandemic [16, 19–21]:

*"Contact tracing seems quite Big Brotherly. I don't think I am willing to submit all my data and all of my contacts for the government to scrutinise who I see regularly. I don't think I will be willing to join the contact tracing apps"*

(Williams 2020; pg. 11)

Participants also mentioned more general privacy and data protection concerns [17, 20, 21], and worries about a contact tracing app not being secure and opening up their phone to hacking [16, 19, 21].

In addition to concerns around privacy associated with a contact tracing app, another emerging theme was the desire to protect 'private' life with contact tracing being seen as an intrusion into that domain [22]:

*"Invasion of privacy—it was not their business to investigate our household"*

(Caleo 2018; pg. 8)

**Mistrust and/or apprehension.** A predominant barrier to engaging with contact tracing was mistrust, of government [16, 20, 22] of contact tracing personnel [23, 24, 26], and of technology. Technology was a particular issue for older people [17, 19].

Participants with experience of contact tracing during the Ebola outbreak, also recollected feelings of apprehension about the behaviours that might be required of them, for example, having to self-isolate and/or refrain from routine activities and worries about the financial and social pressures that this may bring [22, 23].

Some of the mistrust and apprehension around the use of digital contact tracing systems was linked to participants questioning the effectiveness of the system [20, 21, 27]. Participants expressed particular concerns around whether people would be motivated to use a system that did not involve a human interface [27], and if uptake was limited there was the potential for a detrimental effect on the actual (as well as perceived) validity of the system [21].

**Unmet need for more information and support.** Themes of mistrust and apprehension were also associated with gaps in information. These gaps were experienced at the macro level, with participants admitting that they did not understand why contact tracing is needed [17, 20, 21], and questioning whether information could be adequately conveyed through a digital system [27]. Gaps in information provision were also identified at a micro level, with health service staff who had trialled a COVID-19 contact tracing app, highlighting that it was not clear what to do when they were alerted by the app, and there was conflicting advice between the app and a government website [18].

Limited opportunities for support were also identified as potential barriers to engaging with digital contact tracing systems, as participants described how information and warnings may be experienced as more severe because they were not coming from another person, and this also limited the opportunity to provide reassurance and/or support as needed [27].

**Fear of stigmatization.** A salient barrier to the uptake of a contact tracing app was fear of being "branded", concern over the stigmatizing potential of an app were related to concerns around privacy and worries that using the app would enable the identification of individuals with COVID-19 [21]:

*"I actually think that [the contact tracing app] is a terrifying concept. . . it's like being branded with a horrendous black mark. . . . I could look and be like my friend, my neighbour has got Covid."*

(Williams 2020; pg. 13)

There was also a perceived potential for stigmatization associated with being a contact person [21, 26]. Participants described how restrictions imposed through the contact tracing system could ostracize people because they are not able to maintain routine activities [23], and fears that stigmatization around contact tracing could lead to discrimination [21].

**Mode-specific challenges.** In addition to the general and conceptual barriers to engaging with any contact tracing system, as outlined above, there are also mode-specific barriers to

engaging with both digital and manual contact tracing systems, and each mode has its own specific challenges.

Practical and salient barriers to using a contact tracing app included: people not having appropriate devices [17–21]; problems with downloading or installing the app [18]; the impact on other phone applications, and on reducing battery life [18]; and the need for greater app interactivity as participants reported that there were not adequate options for reporting symptoms or test results [18]. In addition to the potential for these actual technical difficulties to create a barrier to uptake of, or engagement with, a contact tracing app, there were also psychological barriers to engaging with technology, as participants lacked confidence in technical proficiency [17, 27], doubted the ease of use [19], felt that it was too much hassle to install [16] and worried about the lack of coordination and oversight with digital systems [27].

Practical barriers were also associated with manual contact tracing systems, as participants described the logistical challenges inherent in identifying contacts [23]; and in recruiting, training and retaining contact tracers [23, 24].

## Discussion

### Principal finding

This review provides strong evidence that many people feel a collective responsibility to help combat infectious disease outbreaks, and that this can be a motivating factor to engage with contact tracing systems. However, engagement with contact tracing relies on these factors outweighing privacy concerns and potential mistrust in the government or public health officials, and on the perception of a personal as well as collective benefit. Providing clear, consistent information and rationale for any contact tracing system, providing support, and emphasising personal as well as collective benefit, is required in order to achieve widespread uptake and engagement.

### Interpretation of results

The review identified a number of higher order conceptual themes that act as facilitators or barriers to any contact tracing system, irrespective of mode of delivery, type of disease (COVID-19 and Ebola), and geographical region. A prominent theme that emerged was the strong sense of collective responsibility felt, and the perception of contact tracing systems as a means of acting on this collective responsibility to help end the infectious disease outbreak and to protect others, particularly those who are vulnerable. Interestingly, this theme was identified across a number of studies conducted in different countries, including the UK which has been perceived as less 'collectivist' than other countries where contact tracing systems have been successfully implemented.

However, although this review reinforced the importance of collective benefit in facilitating engagement with contact tracing, the need for some perceived personal benefit was not entirely superseded, and the evidence suggests that if contact tracing systems can deliver some benefits to the individual as well as the community to which they belong, engagement will be enhanced. The evidence in the review suggests that at the very least individuals should not feel personally disadvantaged from having engaged in contact tracing, and that information and support (including financial support) may help ameliorate this potential barrier. This finding is in line with established behaviour change models, that people need to have sufficient motivation, or health concern to take action [28], and providing clear information on the benefits of engaging and risks of not engaging should be clearly demonstrated so that people know why and how the system will benefit them personally.

Another theme identified, that describes a higher-order facilitator of engagement with contact tracing, is the importance of co-production of contact tracing systems, as demonstrated in the case of Ebola. Evidence synthesised in the review highlights the importance of healthcare systems partnering with communities in order to understand local needs, and how through this process trust and engagement can be cultivated, and effective implementation facilitated.

The need for any contact tracing system to be perceived as efficient, rigorous, and reliable, and the need for information that effectively communicates the purpose of the system as well as the specific actions required, is also highlighted in relation to both digital and manual contact tracing systems. Moreover, the evidence reviewed here suggests that gaps in information and support can foster mistrust and apprehension and act as a salient barrier to engagement. Mistrust of government appeared to arise more in relation to COVID-19, relative to Ebola, however this finding may reflect differences in the modes of delivery with manual contact tracing in Ebola juxtaposed against the COVID-19 studies included here which focused more on app-based contact tracing systems.

The desire to protect 'private' life and the perception of contact tracing as an intrusion into that domain, emerged as a barrier for both manual and digital contact tracing, however, it was particularly prominent in relation to contact tracing applications. Similarly, fear of stigmatization was described in relation to both manual and digital contact tracing, but appeared to be especially pertinent when considering the stigmatizing potential of an app as this was linked to concerns around privacy and data protection. Practical barriers were also identified for both manual and digital contact tracing but seemed particularly relevant to a digital contact tracing system, as a number of real and perceived technical difficulties were highlighted.

## Strengths & limitations

Despite this being a rapid review of the evidence, a comprehensive search was conducted across multiple databases, with screening conducted by two independent reviewers. Moreover, the review includes 12 studies providing a depth of evidence from across the globe, and supporting the generalisability of findings. However, potential limitations include: the possibility that other potentially relevant evidence exists which has not been identified; the moderate to low quality of the included evidence; the inclusion of studies from the pre-print server Medrxiv which have not been scrutinized via the peer review process; and the inclusion of studies that examine intention to engage with a contact tracing system rather than restricting focus to only actual (self-reported or observed) behaviour. Although these potential limitations encourage caution in interpreting the findings, the current relevance and urgency of the review necessitated some pragmatic decisions and the methods adopted helped ensure that the most recent and relevant research was captured.

## Conclusions

Findings from this review suggest that engagement with COVID-19 contact tracing systems could be facilitated by:

1. Clear communication about contact tracing that outlines why a system is needed, how it will work, and highlights personal and community benefit

2. Involvement of stakeholders in the development of contact tracing systems, particularly, digital applications in order to understand and address privacy concerns, the potential for stigmatization, and mistrust and apprehension

3. Evaluation and quality assurance of the contact tracing system in order to create and reinforce the perception of the system as rigorous and reliable

## Supporting information

**S1 Table. Search strategies.**
(DOCX)

**S2 Table. Excluded studies.**
(DOCX)

**S3 Table. Quality appraisal of included studies.**
(DOCX)

**S1 Checklist.**
(DOC)

## Author Contributions

**Conceptualization:** Odette Megnin-Viggars, Patrice Carter, G. James Rubin.

**Formal analysis:** Odette Megnin-Viggars, Patrice Carter.

**Methodology:** Odette Megnin-Viggars, Patrice Carter.

**Supervision:** G. James Rubin.

**Validation:** Odette Megnin-Viggars, Patrice Carter, G. James Rubin.

**Writing – original draft:** Odette Megnin-Viggars.

**Writing – review & editing:** Odette Megnin-Viggars, Patrice Carter, G. J. Melendez-Torres, Dale Weston, G. James Rubin.

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
