## [Decision Letter · Decision Letter 0]

25 Sep 2020

PONE-D-20-28526

Facilitators and barriers to engagement with contact tracing during infectious disease outbreaks: A rapid review of the evidence

PLOS ONE

Dear Dr. Megnin-Viggars,

Thank you for submitting your manuscript to PLOS ONE. After careful consideration, we feel that it has merit but does not fully meet PLOS ONE’s publication criteria as it currently stands. Therefore, we invite you to submit a revised version of the manuscript that addresses the points raised during the review process.

Can you consider the reviewer comments and respond. 

Editor request:

Can you either identify a checklist for reporting rapid reviews (couldn't see one on equator network) OR use reviews such as these to check reporting of methods meets a high standard e.g., https://systematicreviewsjournal.biomedcentral.com/articles/10.1186/s13643-016-0258-9 

https://bmcmedicine.biomedcentral.com/articles/10.1186/s12916-015-0465-6 

We look forward to receiving your revised manuscript.

Kind regards,

Andrew Soundy

Academic Editor

PLOS ONE

Journal Requirements:

2. Please clearly state the exclusion criteria used.

Reviewers' comments:

Reviewer's Responses to Questions

**Comments to the Author**

1. Is the manuscript technically sound, and do the data support the conclusions?

Reviewer #1: Yes

Reviewer #2: Yes

2. Has the statistical analysis been performed appropriately and rigorously? 

Reviewer #1: N/A

Reviewer #2: N/A

3. Have the authors made all data underlying the findings in their manuscript fully available?

Reviewer #1: Yes

Reviewer #2: Yes

4. Is the manuscript presented in an intelligible fashion and written in standard English?

Reviewer #1: Yes

Reviewer #2: Yes

5. Review Comments to the Author

Reviewer #1: This study is a rapid review of contact tracing methods, to identify factors that are associated with greater engagement by patients. The research is interesting because it traces some facilitating elements and obstacles to contact tracing.

The manuscript is written in a clear and concise manner and provides agile information.

The authors refer to the PRISMA statement. However, these criteria apply to systematic reviews. The protocol they developed was not deposited on PROSPERO. Authors should say why they preferred to do a rapid review and which PRISMA criteria they did not follow.

Most of the retrieved studies concern Covid-19 and Ebola. The authors could tell us if there is a difference in the attitude of patients in these two diseases. For example, mistrust seems more frequent in Covid-19.

A few comments on the different geographical origin of the answers could also be interesting

Reviewer #2: This is a nice, concise and topical paper. Methods, recommendations and reporting are clear and relevant flow diagram and checklist for systematic review included. I support publication of this work.

6. PLOS authors have the option to publish the peer review history of their article (what does this mean?). If published, this will include your full peer review and any attached files.

Reviewer #1: **Yes: **Nicola Magnavita

Reviewer #2: No

---

## [Author Response · Author response to Decision Letter 0]

30 Sep 2020

Journal Requirements:

This has been done

2. Please clearly state the exclusion criteria used.

This has been clarified in the manuscript

Reviewer #1: This study is a rapid review of contact tracing methods, to identify factors that are associated with greater engagement by patients. The research is interesting because it traces some facilitating elements and obstacles to contact tracing.

The manuscript is written in a clear and concise manner and provides agile information.

The authors refer to the PRISMA statement. However, these criteria apply to systematic reviews. The protocol they developed was not deposited on PROSPERO. Authors should say why they preferred to do a rapid review and which PRISMA criteria they did not follow. 

Thank you for your comment, this rapid systematic review satisfied all of the PRISMA checklist items, and this has now been clarified in the manuscript. We have also clarified in the manuscript that, unfortunately due to time restrictions, the protocol was not registered on PROSPERO.

Most of the retrieved studies concern Covid-19 and Ebola. The authors could tell us if there is a difference in the attitude of patients in these two diseases. For example, mistrust seems more frequent in Covid-19. Thank you for this interesting observation. 

I think this is true, but interpretation may be confounded by the fact that most of the COVID-19 studies included here focus on app-based contact tracing systems and the Ebola studies on manual contact tracing, so it is possible that it is mode of delivery differences rather than disease differences but I have added some additional text on page 22 of the manuscript to highlight these differences.

A few comments on the different geographical origin of the answers could also be interesting 

I agree this would be interesting, but unfortunately I do not think that the data allows such comparisons given that any geographical differences will be confounded by potential disease differences and mode of delivery differences. However, a prominent finding of this review was that there were a number of higher order conceptual themes that appeared to cut across studies. Previously we had conceptualized these as just cutting across mode of delivery, but I have now added some additional text to page 21 of the manuscript to highlight that these themes emerged irrespective of mode of delivery, type of disease (COVID-19 and Ebola), and geographical region.

Reviewer #2: This is a nice, concise and topical paper. Methods, recommendations and reporting are clear and relevant flow diagram and checklist for systematic review included. I support publication of this work. 

Thank you for your comments

---

## [Decision Letter · Decision Letter 1]

16 Oct 2020

Facilitators and barriers to engagement with contact tracing during infectious disease outbreaks: A rapid review of the evidence

PONE-D-20-28526R1

Dear Dr. Megnin-Viggars,

We’re pleased to inform you that your manuscript has been judged scientifically suitable for publication and will be formally accepted for publication once it meets all outstanding technical requirements.

Kind regards,

Andrew Soundy

Academic Editor

PLOS ONE

Additional Editor Comments (optional):

Reviewers' comments:

Reviewer's Responses to Questions

**Comments to the Author**

1. If the authors have adequately addressed your comments raised in a previous round of review and you feel that this manuscript is now acceptable for publication, you may indicate that here to bypass the “Comments to the Author” section, enter your conflict of interest statement in the “Confidential to Editor” section, and submit your "Accept" recommendation.

Reviewer #1: All comments have been addressed

Reviewer #2: All comments have been addressed

2. Is the manuscript technically sound, and do the data support the conclusions?

Reviewer #1: Yes

Reviewer #2: Yes

3. Has the statistical analysis been performed appropriately and rigorously? 

Reviewer #1: Yes

Reviewer #2: N/A

4. Have the authors made all data underlying the findings in their manuscript fully available?

Reviewer #1: Yes

Reviewer #2: Yes

5. Is the manuscript presented in an intelligible fashion and written in standard English?

Reviewer #1: Yes

Reviewer #2: Yes

6. Review Comments to the Author

Reviewer #1: none. authors have adequately addressed the comments raised in the previous round of review . the language articles must be clear, correct, and unambiguous.

The minimum character count is not useful in this section.

Reviewer #2: (No Response)

7. PLOS authors have the option to publish the peer review history of their article (what does this mean?). If published, this will include your full peer review and any attached files.

Reviewer #1: **Yes: **Nicola Magnavita

Reviewer #2: No

---

## [Editor Report · Acceptance letter]

20 Oct 2020

PONE-D-20-28526R1 

Facilitators and barriers to engagement with contact tracing during infectious disease outbreaks: A rapid review of the evidence 

Dear Dr. Megnin-Viggars:

I'm pleased to inform you that your manuscript has been deemed suitable for publication in PLOS ONE. Congratulations! Your manuscript is now with our production department. 

Kind regards, 

on behalf of

Dr. Andrew Soundy 

Academic Editor

PLOS ONE